# Associations of ultrasound-based inflammation patterns with peripheral innate lymphoid cell populations, serum cytokines/chemokines, and treatment response to methotrexate in rheumatoid arthritis and spondyloarthritis

**Manami Kato**[1], **Kei Ikeda**[1]*, **Takahiro Sugiyama**[1], **Shigeru Tanaka**[1], **Kazuma Iida**[1], **Kensuke Suga**[1], **Nozomi Nishimura**[1], **Norihiro Mimura**[1], **Tadamichi Kasuya**[1], **Takashi Kumagai**[1], **Hiroki Furuya**[1], **Taro Iwamoto**[1], **Arifumi Iwata**[1], **Shunsuke Furuta**[1], **Akira Suto**[1], **Kotaro Suzuki**[1], **Eiryo Kawakami**[2,3], **Hiroshi Nakajima**[1]

1 Department of Allergy and Clinical Immunology, Graduate School of Medicine, Chiba University, Chiba, Chiba, Japan, 2 Artificial Intelligence Medicine, Graduate School of Medicine, Chiba University, Chiba, Chiba, Japan, 3 Medical Sciences Innovation Hub Program, RIKEN, Wako, Saitama, Japan

* K.Ikeda@faculty.chiba-u.jp

## Abstract

### Objectives

We aimed to explore the associations of musculoskeletal inflammation patterns with peripheral blood innate lymphoid cell (ILC) populations, serum cytokines/chemokines, and treatment response to methotrexate in patients with rheumatoid arthritis (RA) and spondyloarthritis (SpA).

### Methods

We enrolled 100 patients with either RA or SpA and performed ultrasound to evaluate power Doppler signals for synovitis (52 joint regions), tenosynovitis (20 tendons), and enthesitis (44 sites). We performed clustering analysis using unsupervised random forest based on the multi-axis ultrasound information and classified the patients into groups. We identified and counted ILC1-3 populations in the peripheral blood by flow cytometry and also measured the serum levels of 20 cytokines/chemokines. We also determined ACR20 response at 3 months in 38 patients who began treatment with methotrexate after study assessment.

### Results

Synovitis was more prevalent and severe in RA than in SpA, whereas tenosynovitis and enthesitis were comparable between RA and SpA. Patients were classified into two groups which represented synovitis-dominant and synovitis-nondominant inflammation patterns. While peripheral ILC counts were not significantly different between RA and SpA, they were

**Data Availability Statement:** All relevant data are within the paper and its Supporting Information files.

**Funding:** This work was supported by Grants-in-Aids for Scientific Research from the Ministry of Education, Culture, Sports, Science and Technology (MEXT) (https://www.mext.go.jp/en/) (Grant number: 18K08405). The funders had no role in study design, data collection and analysis, decision to publish, or preparation of the manuscript.

**Competing interests:** The authors declare that they have no competing interests as regards this work.

significantly higher in the synovitis-nondominant group than in the synovitis-dominant group (ILC1-3: $p = 0.0007$, $p = 0.0061$, and $p = 0.0002$, respectively). On the other hand, clustering of patients based on serum cytokines/chemokines did not clearly correspond either to clinical diagnoses or to synovitis-dominant/nondominant patterns. The synovitis-dominant pattern was the most significant factor that predicted clinical response to methotrexate ($p = 0.0065$).

## Conclusions

Musculoskeletal inflammation patterns determined by ultrasound are associated with peripheral ILC counts and could predict treatment response to methotrexate.

## Introduction

Rheumatoid arthritis (RA) and spondyloarthritis (SpA) are the most archetypal entities of inflammatory arthritis. The clinical feature of RA has long been characterized by arthritis, which is the clinical manifestation of synovitis [1], while tenosynovitis has also been shown to be an important pathologic feature of RA that is significantly associated with disease and structural progression [2, 3]. On the other hand, musculoskeletal manifestations of SpA, such as psoriatic arthritis (PsA) and ankylosing spondylitis (AS), are more heterogeneous than those of RA, which encompass arthritis, enthesitis, dactylitis, and axial disease [4–6]. Among these, however, enthesitis has been considered the primary site of inflammation, which causes synovitis, tenosynovitis, and soft-tissue inflammation, resulting in a variety of clinical manifestations [7, 8]. It is clinically important to distinguish between these types of musculoskeletal manifestations in SpA because treatment response to medication such as methotrexate (MTX) can be different between the arthritis-dominant disease and the others [9–12]. These data also suggest that the type of musculoskeletal manifestation represents underlying cellular/molecular pathways in SpA; whereas whether the same concept applies to RA remains unknown.

Imaging modalities, such as ultrasound and magnetic resonance imaging (MRI), have played significant roles in understanding the pathophysiology of musculoskeletal inflammation in RA and SpA [13–15]. Importantly, ultrasound and MRI have demonstrated better accuracy in detecting and locating the inflammation than physical examination [13, 16–18]. However, the different types of musculoskeletal inflammation have not been thoroughly assessed using these modern imaging techniques for both RA and SpA in a uniform study setting. Therefore, neither clustering of these patients based on comprehensive imaging data nor analyses of associations between such clusters and cellular/molecular biomarkers have been reported to date.

Innate lymphoid cells (ILCs) are the effector cells of the innate immune system, which are distinct from conventional lymphocytes and involved in protection against pathogens, tissue remodeling, and maintenance of homeostasis [19]. Three ILC subsets, Group 1–3 ILC (ILC1-3), have been identified based on the key transcription factors and cytokine production that resemble helper T cell counterparts [20]. The numbers of ILC1 and ILC3 were found to be increased in the peripheral blood, synovial fluid, and lymph node of patients with RA and juvenile idiopathic arthritis [21, 22], whereas the number of peripheral ILC2 was decreased [23]. On the other hand, ILC3 has been shown to be present in the healthy entheses, to expand in the peripheral blood, gut, synovial fluid, or bone marrow in SpA patients, and to produce key SpA-related inflammatory cytokines [24–29]. Although these reports indicate that ILCs

are involved in the pathophysiology of RA and SpA, the associations between ILCs and the patterns of musculoskeletal inflammation are unknown. Similarly, while a number of studies have compared serum cytokines/chemokines between RA, PsA, and controls [30], their associations with musculoskeletal inflammation patterns are poorly understood.

We hypothesized that the patterns of musculoskeletal inflammation determined by comprehensive ultrasound assessment better represent the underlying cellular and molecular pathways and are better predictors of treatment response than clinical manifestations. We also hypothesized that this association between local inflammation patterns and cellular/molecular pathways is present not only in SpA but also in RA. On the basis of these hypotheses, in this study, we aimed to explore the associations of ultrasound-based inflammation patterns with peripheral blood ILC populations, serum cytokines/chemokines, and treatment response to MTX in patients with RA and SpA.

## Materials and methods

### Patients

We screened consecutive patients with either RA or SpA including PsA who had peripheral symptoms upon a visit at the outpatient clinic of Department of Allergy and Clinical Immunology between August 2018 and February 2020. Inclusion criteria for recruitment were: 18 years or older, fulfilled either 2010 American College of Rheumatology (ACR)/ European League Against Rheumatism (EULAR) RA Classification Criteria [1], Classification Criteria for Psoriatic Arthritis (CASPAR) for PsA [4], or Assessment of SpondyloArthritis international Society (ASAS) Criteria for axial or peripheral SpA [5, 6], had any peripheral musculoskeletal symptom (i.e. pain, swelling, and/or stiffness in the joint, enthesis, finger, and/or toe), and had disease duration of 10 years or less. Patients who had a concomitant systemic autoimmune/ inflammatory disease other than Sjogren syndrome were excluded. We enrolled patients when they gave written informed consent to participate the study, and patients did not receive any study procedures before giving it.

All study procedures were approved by Chiba University Ethics Committee (approved number: 3121) and performed in accordance with the Declaration of Helsinki. The study has been registered in UMIN Clinical Trial Registry (ID: 000033797).

### Clinical assessment

Patients underwent thorough history taking, chart reviewing, and physical assessment. Specific assessment for RA and SpA included full swollen/tender joint counts (66/68 joints) [31], Spondyloarthritis Research Consortium of Canada (SPARCC) Enthesis Index [32], dactylitis count (0–20), patient's pain Visual Analogue Scale (VAS), patient's global assessment VAS, physician's global assessment VAS, and Health Assessment Questionnaire-Disability Index (HAQ-DI). Peripheral white blood cell counts and serum levels of C-reactive protein were obtained on the same day of clinical assessment, whereas information on anti-citrullinated protein antibody (ACPA)/rheumatoid-factor (RF) positivity was collected by chart reviewing. We also collected information on medications that patients were receiving for RA or SpA at the time of study assessment.

### Ultrasound examination

We performed ultrasound assessment for synovitis, tenosynovitis, and enthesitis on the same day of clinical assessment using either a HI VISION Avius or a HI VISION Ascendus with a linear multifrequency (5-18MHz) probe (Hitachi Medical Corporation, Tokyo, Japan)

depending on availability. Ultrasound was performed by either one of the 10 trained sonographers (MK, KIkeda, TS, KIida, KSuga, NN, NM, TKasuya, TKumagai, SF) who were instructed not to see the patient's clinical information. For synovitis, we scanned 26 joint sites bilaterally: 1st interphalangeal (IP) joint, 2nd-5th proximal interphalangeal (PIP) joints, 1st-5th metacarpophalangeal (MCP) joints, radiocarpal joint, midcarpal joint, distal radioulnar joint, humeroradial joint, humeroulnar joint, olecranon fossa, suprapatellar pouch, medial aspect of knee, lateral aspect of knee, tibiotalar joint, talonavicular joint, and 1st-5th metatarsophalangeal (MTP) joints. We excluded distal interphalangeal (DIP) joints, in which discriminating between synovial and entheseal power Doppler (PD) signals is particularly difficult. We also excluded shoulder, where intra-articular synovium and tenosynovium are continuous and inseparable, and degenerative lesions are prevalent. We scored synovial PD signals in each joint region semiquantitatively (0–3) (synovitis score) with a previously validated method [13, 18, 33].

For tenosynovitis, we scanned 10 tendons bilaterally: flexor digitorum tendons of 1st-5th fingers, 2nd, 4th, and 6th compartments of wrist extensor tendons, tibialis posterior, and peroneal tendons. We scored tenosynovial PD signals in each tendon region semiquantitatively (0–3) (tenosynovitis score) with a previously validated method [13, 34].

For enthesitis, we scanned 12 entheseal sites bilaterally: extensor digitorum insertions into 1st-5th proximal phalanx, lateral epicondyle, medial epicondyle, triceps insertion at olecranon, quadriceps insertion at patella, patellar tendon insertions at patella and tibia, and Achilles tendon insertion. We also scanned the extensor digitorum tendons at 1st-5th metacarpal heads and 1st-5th A1 pulleys, which are considered "functional entheses", where inflammation equivalent to enthesitis occurs in PsA/SpA [7, 35]. We scored entheseal PD signals in each tendon region semiquantitatively (0–3) (enthesitis score) based on sonographer's subjective assessment (0, none; 1, mild signals; 2, moderate signals; 3, severe signals).

We calculated total synovitis/tenosynovitis/enthesitis scores by summating all synovitis/tenosynovitis/enthesitis scores in each patient.

## ILC counts

On the same day of clinical and ultrasound assessment, we obtained fresh whole blood, isolated peripheral blood mononuclear cells (PBMCs) by Ficoll-Paque Premium (GE Healthcare, Chicago, IL, USA), and stored them in CELLBANKER 2 freezing medium (Nippon Zenyaku Kogyo, Fukushima, Japan) at –80°C. We analyzed the samples by flow cytometry using the following antibodies for surface staining: FITC-conjugated anti-CD3 (UCHT1), anti-CD14 (HCD14), anti-CD19 (HIB19), anti-CD11c (3.9), anti-FcεRIα (AER-37 [CRA-1]), anti-CD94 (DX22), anti-CD123 (6H6), anti-CD34 (581), anti-CD16 (3G8), BV510-conjugated anti-CD45 (HI30), BV421-conjugated anti-CD127 (A019D5), PE-Cy7-conjugated anti-CD117 (c-kit) (104D2), Alexa Fluor 647-conjugated anti-CRTH2 (BM16), and APC-Cy7-conjugated anti-CD56 (HCD56) (All antibodies were from BioLegend, San Diego, CA, USA).

Data were acquired on FACS CANTO II (BD Biosciences, San Jose, CA, USA) and analyzed with FlowJo software (Tree Star, Ashland, OR, USA). Total ILC, ILC1, ILC2, and ILC3 were identified using previously reported gating methods [36] (S1 Fig).

We confirmed no significant differences in any populations tested before and after freezing.

## Serum cytokines/chemokines

On the same day of clinical and ultrasound assessment, we obtained and stored the patients' sera at –80°C. On the basis of previous reports [30, 37, 38], we measured serum levels of β-defensin 2 and calprotectin using ELISA kits (Arigo, Hsinchu, Taiwan; Hycult Biotech, Uden,

The Netherlands, respectively). We also measured serum levels of CCL20/MIP3a, GM-CSF, IFN-γ, IL-1β, IL-6, IL-8, IL-9, IL-10, IL-12p70, IL-15, IL-17A, IL-17F, IL-21, IL-22, IL-23, Lipocalin-2/NGAL, and TNF-α using MILLIPLEX Assay Kits and a MAGPIX Multiplexing System (Merck Millipore, Darmstadt, Germany). Data were analyzed using xPONENT 4.2 Software (Luminex Corporation, Austin, TX, USA) and the average of duplicate samples was calculated.

One RA patient with a result of extremely high values for most of the cytokines/chemokines was excluded from cytokine/chemokine analyses as an outlier due to measurement error.

### Treatment response

We evaluated treatment response in patients who received a disease modifying anti-rheumatic drug (DMARD) as their initial treatment within 12 weeks after study assessment and continued treatment for 12 weeks or longer. Treatment response was assessed by the ACR 20/50/70% improvement criteria (ACR20/50/70) [39] at 12 weeks.

### Statistics

Given the exploratory nature of this study, we did not perform sample size or power calculation.

Normally distributed continuous data were summarized with mean and standard deviation (SD) and were compared using Welch's T test. Non-normally distributed data were summarized with median and interquartile range (IQR) and analyzed using Brunner-Munzel test and Spearman's correlation coefficient. Categorical data were presented with numbers and percentages and compared using Chi's square or Fisher's exact test.

Hierarchical clustering was performed to analyze the similarity among patients based on serum cytokine/chemokine levels. An unsupervised random forest dissimilarity measure [40, 41] was used to evaluate the similarity among patients based on ultrasound scores. The random forest dissimilarity was used as input for Uniform Manifold Approximation and Projection (UMAP) [42], which provides a visual representation of the positional relationship among a set of patients. Subsequently, Partitioning Around Medoids (PAM) clustering [43] was applied on the two scaling coordinates of UMAP.

Welch's t-test, Chi's square test, Fisher's exact test, Spearman's correlation coefficient, and hierarchical clustering were performed using JMP Pro 13.2 (SAS Institute Inc., Cary, NC, USA). Brunner-Munzel test was performed using R lawstat package. Random forest, UMAP, and PAM clustering were performed using python scikit-learn package, python umap package, and R cluster package, respectively. Two-sided p-values less than 0.05 were considered statistically significant. Due to the exploratory nature of this study, we did not correct p-values for multiple testing.

## Results

### Patients

A total of 100 patients were enrolled (RA 66, SpA 34 [PsA 29, non-PsA 5]). Table 1 summarizes the characteristics of the patients and disease, as well as the treatment. Among SpA patients, six and one patients fulfilled the ASAS classification of radiographic and non-radiographic axial SpA, respectively [5]; four had or previously had inflammatory back pain but were not classified as axial SpA. All PsA patients had or previously had psoriatic skin lesions. Two non-PsA SpA patients had ulcerative colitis and one PsA patient had uveitis. One RA patient had Sjogren's syndrome and three had local autoimmune diseases (Type 1 diabetes

**Table 1. Patient and disease characteristics.**

| | Total (n = 100) | Rheumatoid arthritis (n = 66) | Spondyloarthritis (n = 34) |
|---|---|---|---|
| Mean age ± SD, year-old | 57.8 ± 14.6 | 61.5 ± 14.9 | 50.7 ± 11.2 |
| Female, number (%) | 60 (60.0) | 50 (75.8) | 10 (29.4) |
| Mean Body Mass Index ± SD, kg/m$^2$ | 23.5 ± 4.01 | 22.4 ± 3.20 | 25.8 ± 4.49 |
| Mean disease duration ± SD, month | 32.1 ± 36.1 | 28.5 ± 34.4 | 39.1 ± 38.7 |
| Median swollen joint count (66 joint) (IQR) | 3 (1–7) | 5 (2–10.25) | 1 (0–2.25) |
| Median tender joint count (68 joint) (IQR) | 3 (1–5.75) | 3 (1–7) | 2 (1–5) |
| Median SPARCC Enthesitis Index (0–16) (IQR) | 0 (0–0.75) | 0 (0–0) | 1 (0–2) |
| Median dactylitis score (20 finger/toe) (IQR) | 0 (0–0) | 0 (0–0) | 0 (0–0) |
| Median HAQ-DI (IQR) | 0.5 (0.125–1) | 0.6875 (0.125–1.125) | 0.25 (0–0.906) |
| Median serum C-reactive protein (IQR), mg/dl | 0.30 (0.08–1.76) | 0.27 (0.07–1.92) | 0.34 (0.09–1.47) |
| Mean Patient's Pain VAS (0–100) ± SD | 43.2 ± 27.2 | 46.1 ± 26.2 | 37.6 ± 28.8 |
| Mean Patient's Global VAS (0–100) ± SD | 49.4 ± 29.6 | 52.6 ± 28.9 | 43.1 ± 30.3 |
| Mean Physician's Global VAS (0–100) ± SD | 35.8 ± 24.7 | 41.9 ± 23.8 | 24.0 ± 22.2 |
| Rheumatoid factor positive, number (%) | 52 (52.0) | 49 (74.2) | 3 (8.8) |
| Anti-citrullinated protein antibody positive, number (%) | 48 (48.0) | 48 (72.7) | 0 (0.0) |
| Inflammatory back pain ever present, number (%) | 11 (11.0) | 0 (0.0) | 11 (32.4) |
| Psoriatic skin lesion ever present, number (%) | 29 (29.0) | 0 (0.0) | 29 (85.3) |
| Nail lesion ever present, number (%) | 19 (19.0) | 0 (0.0) | 19 (55.9) |
| No treatment, number (%) | 23 (23.0) | 14 (21.2) | 9 (26.5) |
| Receiving NSAID, number (%) | 46 (46.0) | 33 (50.0) | 13 (38.2) |
| Receiving Glucocorticoid, number (%) | 3 (3.0) | 2 (3.0) | 1 (3.0) |
| Receiving csDMARD, number (%) | 32 (32.0) | 23 (34.8) | 9 (26.5) |
| Receiving bDMARD, number (%) | 9 (9.0) | 6 (9.1) | 3 (8.8) |
| Receiving tsDMARD, number (%) | 5 (5.0) | 1 (1.5) | 4 (11.8) |

SD: standard deviation; IQR: interquartile range; SPARCC: Spondyloarthritis Research Consortium of Canada; HAQ-DI: Health Assessment Questionnaire-Damage Index; VAS: visual analogue scale; NSAID: non-steroidal anti-inflammatory drug; DMARD: disease-modifying anti-rheumatic drug; csDMARD: conventional synthetic DMARD; bDMARD: biological DMARD; tsDMARD: targeted synthetic DMARD.

mellitus 2 [RA 1, PsA 1]; Basedow's disease 1 [PsA]). Patient's and Physician's Global VAS indicated that the study subjects largely represented patients with mild to moderate arthritic disease activity.

## Ultrasound and clustering

Fig 1 summarizes the ultrasound results. As expected, active synovitis was more prevalent in RA than in SpA in the vast majority of joints assessed, with MCP and wrist joints being the most commonly inflamed joints (Fig 1A). On the other hand, the prevalence of active tenosynovitis was similar between RA and SpA overall although it was higher in 4$^{th}$ and 6$^{th}$ compartments of wrist extensor tendons in RA (Fig 1B). Unexpectedly, the prevalence of active enthesitis was not higher in SpA overall and it was even higher in the functional entheses of extensor digitorum in RA (Fig 1C). Consistently, total synovitis score was significantly higher in RA than in SpA, while total tenosynovitis score and total enthesitis score were comparable between RA and SpA (S2 Fig).

Next, we assessed the similarity among these patients based on these ultrasound scores using unsupervised random forest and UMAP as described in the Patients and Methods section. As shown in Fig 2A, patients are clustered into two groups (Group 1, n = 38; Group 2,

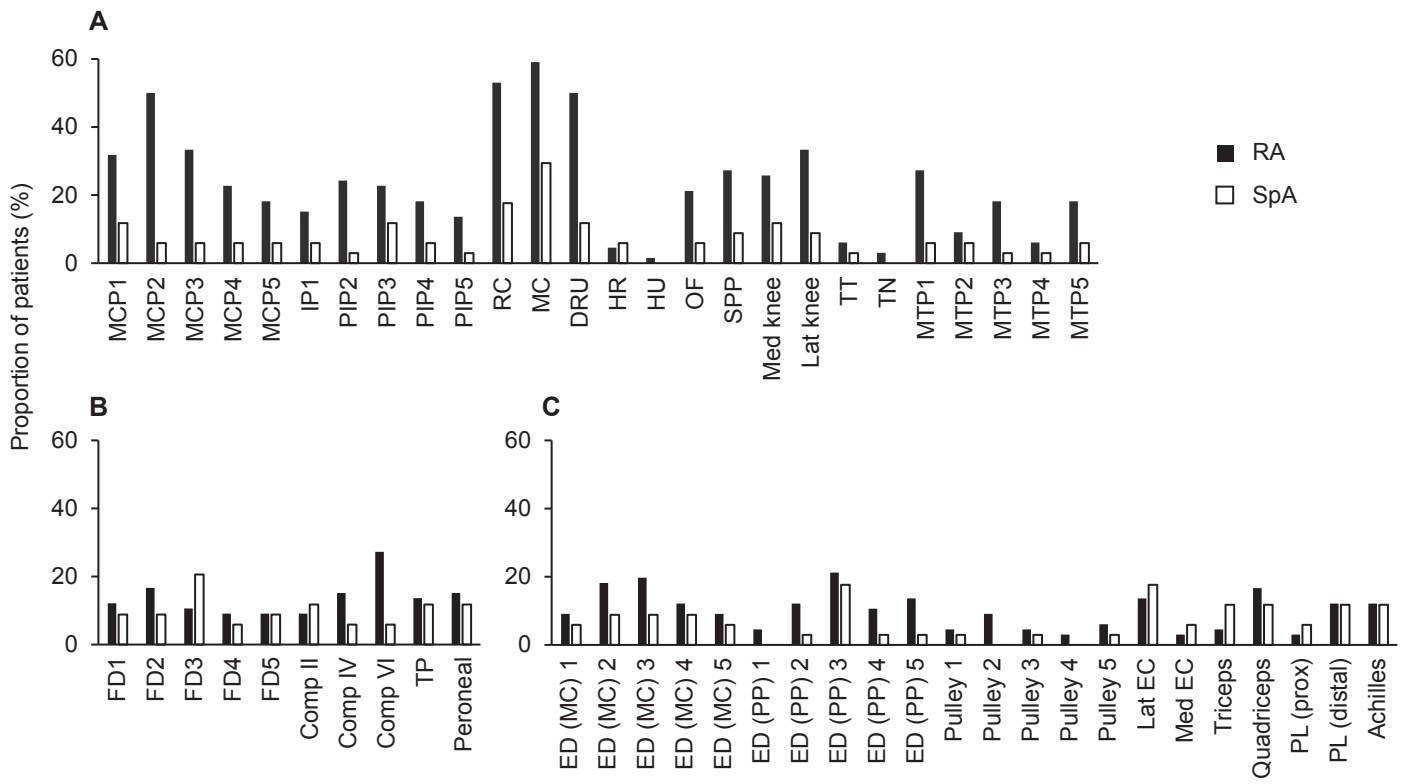

**Fig 1. Proportions of patients with positive power Doppler score for each site scanned.** (A) Joints evaluated for synovitis. (B) Tendons evaluated for tenosynovitis. (C) Entheses evaluated for enthesitis. RA: rheumatoid arthritis; SpA: spondyloarthritis; MCP: metacarpophalangeal; IP: interphalangeal; PIP: proximal interphalangeal; RC: radiocarpal; MC: midcarpal; DRU: distal radioulnar; HR: humeroradial; HU: humeroulnar; OF: olecranon fossa; SPP: suprapatellar pouch; Med knee: medial aspect of knee; Lat knee: lateral aspect of knee; TT: tibiotalar; TN: talonavicular; MTP: metatarsophalangeal; FD: flexor digitorum; Comp: compartment of wrist extensor tendon; TP: tibialis posterior; ED (MC): extensor digitorum at metacarpal head; ED (PP): extensor digitorum insertion into proximal phalanx; Lat EC: lateral epicondyle; Med EC: medial epicondyle; PL (prox): proximal attachment of patellar ligament; PL (distal): distal attachment of patellar ligament.

n = 62). Interestingly, this ultrasound-based grouping did not clearly correspond to patients' diagnosis or ACPA positivity although the majority of PsA patients belonged to Group 2 (Fig 2B and 2C). On the other hand, patients with a high total synovitis score clustered into Group 1, whereas those with a low synovitis score clustered into Group 2 (Fig 2D). Patients with a high total tenosynovitis score tended to be clustered into Group 1 (Fig 2E), while those with a high enthesitis score tended to cluster into Group 2 (Fig 2F). We did not find any significant associations between this grouping and the location of inflamed joint regions (e.g. small vs. large joints, fingers/toes vs. others, upper vs. lower extremities). These data demonstrate that Group 1 represents patients with a "synovitis-dominant" inflammation pattern, whereas Group 2 represents those with a "synovitis-nondominant" inflammation pattern.

## ILC

There were no significant differences between RA and SpA in either white blood cell count, lymphocyte count, or neutrophil count. As shown in Fig 3A, median numbers of ILCs were numerically larger in SpA than in RA but the differences were, unexpectedly, not statistically significant.

On the other hand, when mapped on ultrasound-based clustering, the numbers of ILCs, particularly ILC3, seemed to be higher in Group 2 than in Group 1 (Fig 2G and 2H). Indeed,

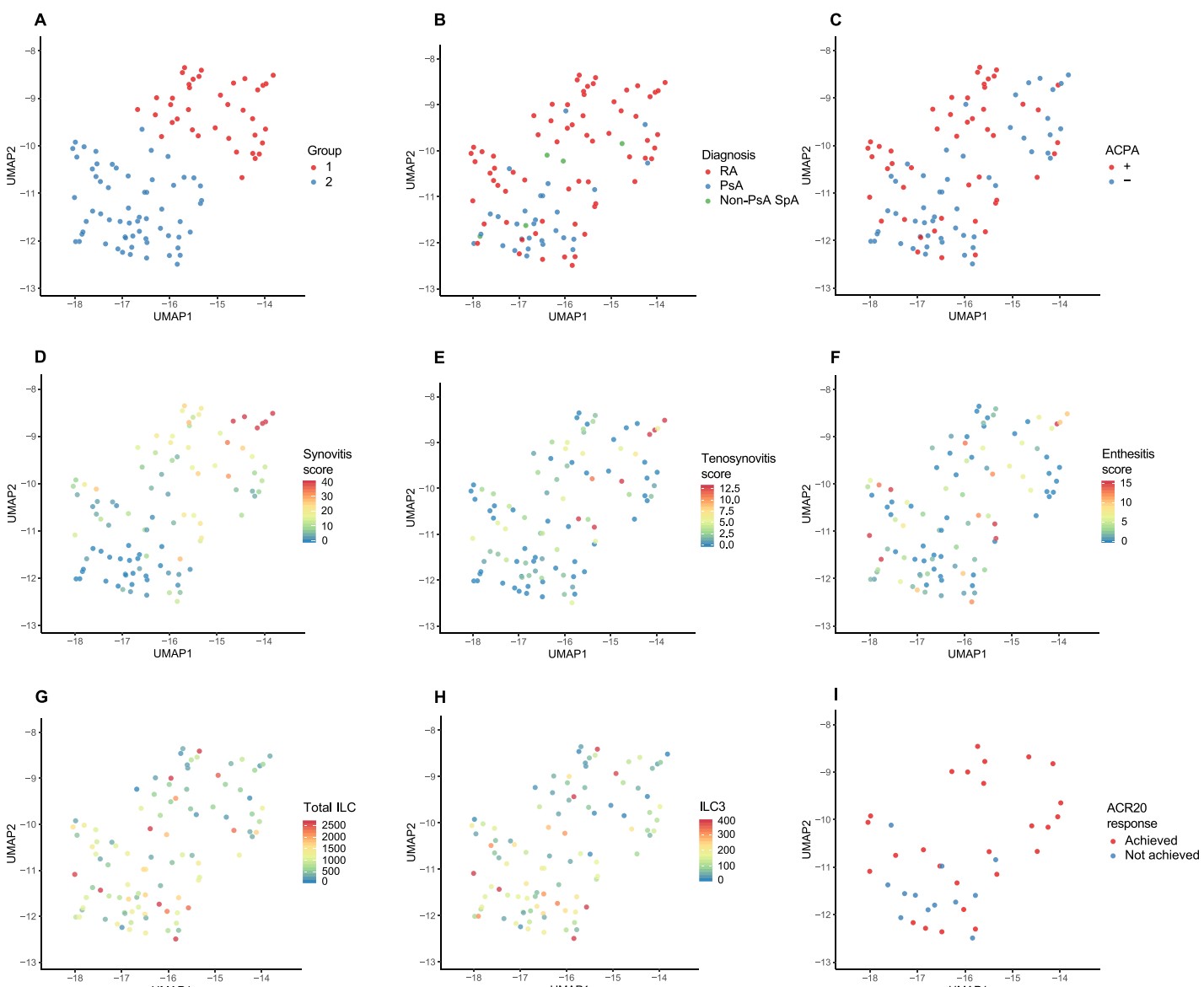

**Fig 2. Clustering of patients based on ultrasound assessment.** Positional relationship among patients (n = 100) based on power Doppler scores (116 sites) is plotted by Uniform Manifold Approximation and Projection (UMAP). (A) Partitioning Around Medoids (PAM) clustering of patients based on the two scaling coordinates of UMAP. (B-I) Parameters are mapped on the UMAP plot: (B) clinical diagnoses (rheumatoid arthritis [RA], psoriatic arthritis [PsA], non-PsA spondyloarthritis [SpA]), (C) anti-citrullinated protein antibody (ACPA) positivity, (D) total synovitis score, (E) total tenosynovitis score, (F) total enthesitis score, (G) peripheral count of total innate lymphoid cell (ILC), (H) peripheral ILC3 count, (I) ACR20 response to methotrexate (MTX).

median ILC counts were significantly higher in Group 2 than in Group 1 (Fig 3B), indicating that peripheral ILC numbers are more closely associated with musculoskeletal inflammation patterns than with clinical diagnoses.

## Serum cytokines/chemokines

There were significant differences found between RA and SpA in serum levels of β-defensin 2, IFN-γ, IL-22, IL-23, IL-17F, GM-CSF, CCL20/MIP3a, IL-15, IL-1β, IL-17A, and IL-6 (S1 Table). However, hierarchical clustering demonstrated that the cytokine/chemokine profile as a whole does not clearly discriminate between RA and SpA (Fig 4).

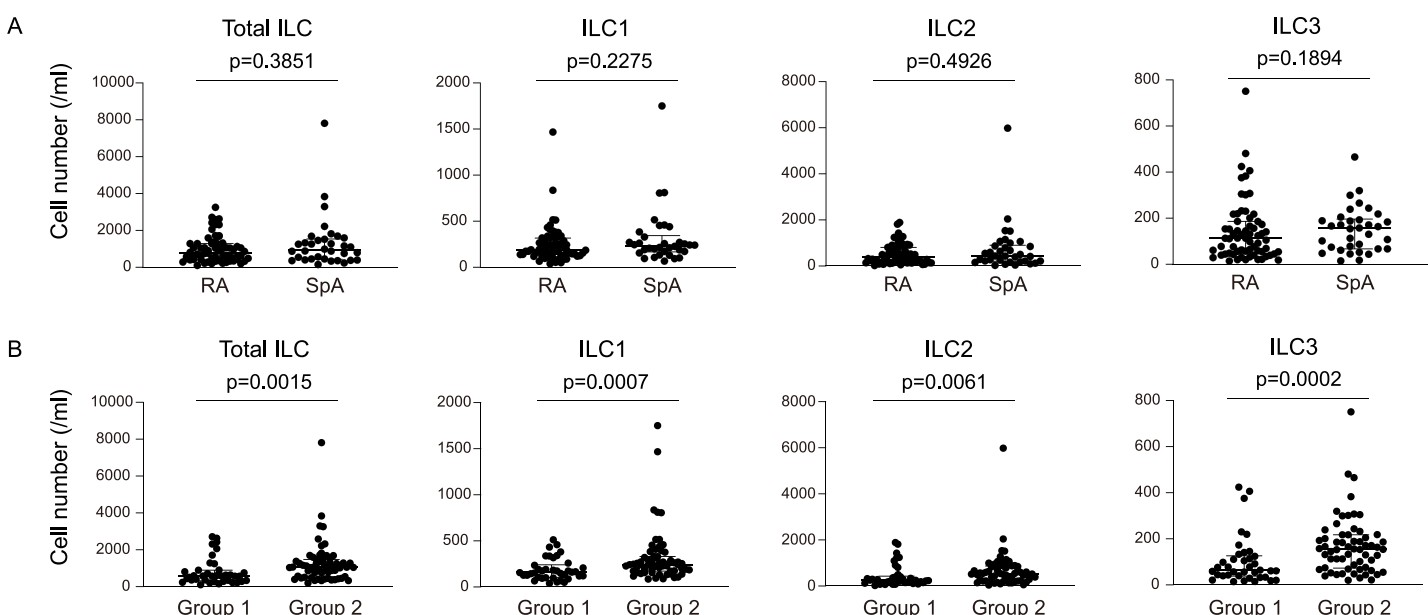

**Fig 3. Differences in innate lymphoid cell numbers between groups.** Numbers of total innate lymphoid cells (ILCs) and Group 1–3 ILCs (ILC1, ILC2, and ILC3) in the peripheral blood are compared (A) between patients with RA and those with SpA and (B) between Group 1 (synovitis-dominant pattern) and Group 2 (synovitis-nondominant pattern).

There were also significant differences between Group 1 and Group 2 in the serum levels of C-reactive protein, calprotectin, β-defensin 2, and CCL20/MIP3a (S2 Table). Interestingly, β-defensin 2 was the only serum protein that was significantly different both between RA and SpA and between Group 1 and Group 2. Nevertheless, the cytokine/chemokine clustering also did not clearly discriminate between Group 1 and Group 2 (Fig 4).

We also analyzed correlations between serum cytokines/chemokines and peripheral ILC numbers. Only a few combinations showed significant correlations although these correlations were all weak (S3 Table).

## Treatment response

Thirty-eight patients (RA 26, SpA 12) initiated MTX-monotherapy (n = 37) or added MTX to salazosulfapyridine (n = 1) within 12 weeks of study assessment. All of these patients stayed on the same regimen for 12 weeks or longer and the median (IQR) maximum weekly dose of MTX was 10 (10–12) mg. At 12 weeks, 26 (68.4%), 21 (55.3%), and 17 (44.7%) patients achieved ACR20, ACR50, and ACR70 responses, respectively.

Table 2 summarizes the differences in baseline factors between patients who achieved ACR20 response to MTX and those who did not. The most significant factor that predicted treatment response was the Group 1 inflammation pattern, which also predicted ACR50 (p = 0.0336) and ACR70 (p = 0.0159). Indeed, all patients in Group 1 achieved ACR20 to MTX at 12 weeks (Fig 2I). The other significant predictive factors for treatment response to MTX were older age, no psoriatic skin lesion, diagnosis of RA, no inflammatory back pain, and higher swollen joint count (Table 2).

## Discussion

Here, we demonstrate that patients with RA and SpA can be classified into two groups based on the anatomical locations of musculoskeletal inflammation determined by ultrasound

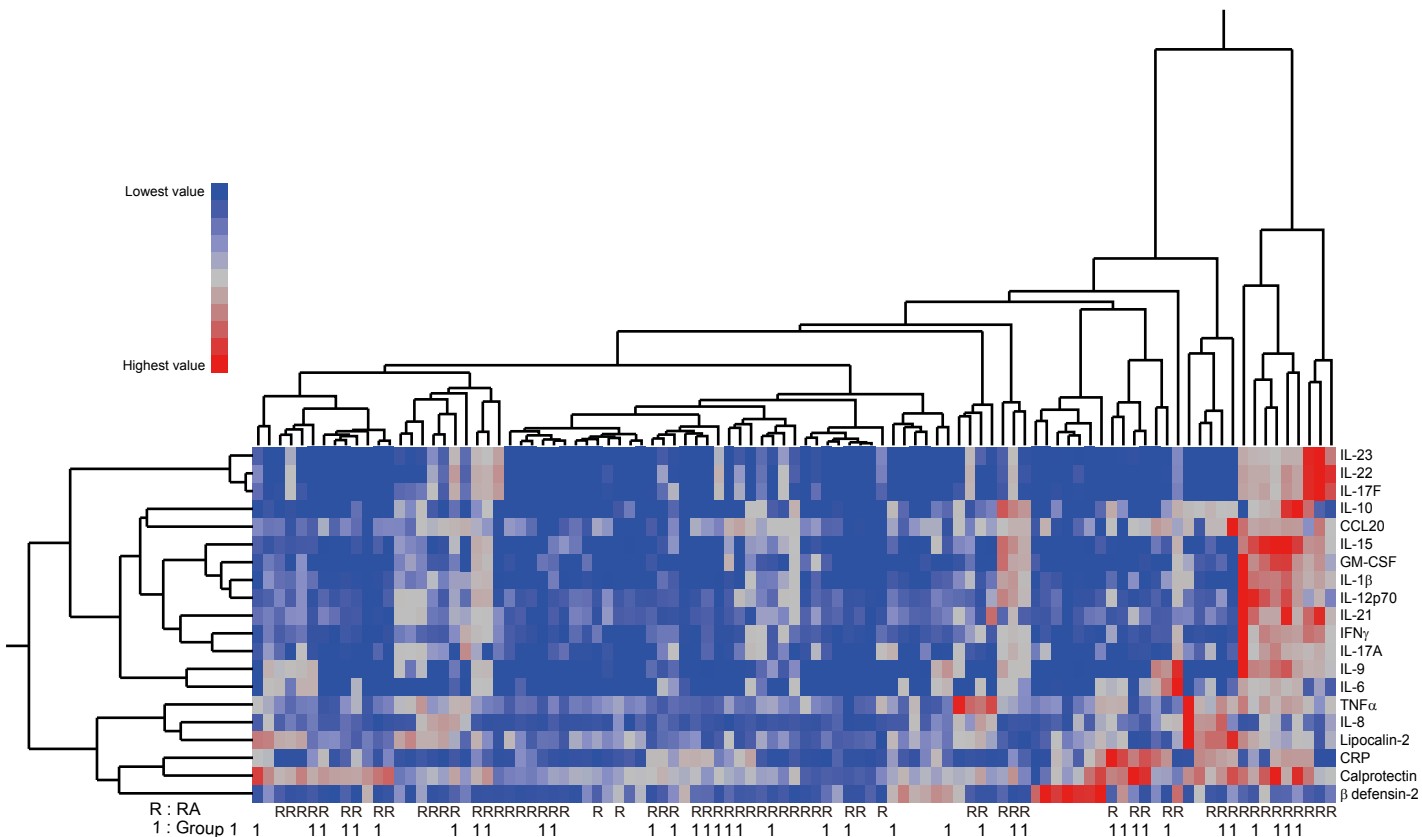

**Fig 4. Clustering of patients based on serum cytokines/chemokines.** A heatmap representing the serum levels of 20 biomarkers and dendrograms illustrating hierarchical clustering of these cytokines/chemokines and patients (n = 99). The clinical diagnosis (rheumatoid arthritis vs. spondyloarthritis) and the ultrasound-based inflammation pattern (Group 1 vs. Group 2) are provided below the heatmap.

examination of peripheral joints, tendons, and entheses. These clusters represented "synovitis-dominant" and "synovitis-nondominant" patterns but did not clearly correspond to clinical diagnoses (i.e., RA and SpA). Furthermore, peripheral ILC counts were significantly associated with ultrasound-based inflammation patterns but not with clinical diagnoses. In addition, the synovitis-dominant pattern was the strongest predictor of clinical response to MTX.

To our knowledge, this is the first comprehensive ultrasound assessment for synovitis, tenosynovitis, and enthesitis in both RA and SpA within the same research setting. The most unexpected finding was that RA patients showed enthesitis as frequently as SpA patients (Fig 1). In particular, the inflammation in extensor digitorum at metacarpal head (i.e., functional enthesis), which was reported to distinguish between PsA and RA [44], was identified more frequently in RA than in SpA. Although the seemingly lower disease activity in SpA patients and the possible influence from adjacent synovitis need to be taken into consideration, our data indicate that a significant proportion of RA patients develop enthesitis. Together with the already known notion that the peripheral musculoskeletal inflammation in SpA is diverse, our data suggest that musculoskeletal inflammation in RA and SpA are considerably heterogeneous, consisting of different ratios of synovitis, tenosynovitis, and enthesitis in individual patients.

This study is also the first to classify patients with inflammatory arthritis using a clustering method incorporating multi-axis ultrasound information. The clustering demonstrates that these patients can be divided into two groups depending on the predominance of synovitis

**Table 2. Differences in baseline variables between patients who responded to methotrexate and those who did not.**

| | ACR20 achieved (n = 26) | ACR20 not achieved (n = 12) | p-value |
|---|---|---|---|
| Diagnosis: rheumatoid arthritis, number (%) | 21 (80.8) | 5 (41.7) | 0.0258 |
| Mean age ± SD, year-old | 61.6±14.9 | 50.1±13.3 | 0.0252 |
| Female, number (%) | 15 (57.7) | 7 (58.3) | 1.0000 |
| Mean Body Mass Index ± SD, kg/m$^2$ | 23.54±3.70 | 23.90±4.24 | 0.7998 |
| Mean disease duration ± SD, month | 11.0±12.7 | 14.4±11.8 | 0.4216 |
| Median swollen joint count (66 joint) (IQR) | 5 (3–16) | 2 (1–6.75) | 0.0300 |
| Median tender joint count (68 joint) (IQR) | 4.5 (2.75–8.25) | 3 (1–6) | 0.1144 |
| Median SPARCC Enthesitis Index (0–16) (IQR) | 0 (0–0.25) | 0.5 (0–2) | 0.1455 |
| Median dactylitis score (20 finger/toe) (IQR) | 0 (0–0) | 0 (0–1) | 0.0512 |
| Median HAQ-DI (IQR) | 0.75 (0.25–1.16) | 0.5625 (0.031–1) | 0.2485 |
| Median serum C-reactive protein (IQR), mg/dl | 0.63 (0.21–3.22) | 0.265 (0.09–3.84) | 0.7467 |
| Mean Patient's Pain VAS (0–100) ± SD | 54.7±27.7 | 43.2±25.1 | 0.2146 |
| Mean Patient's Global VAS (0–100) ± SD | 56.1±30.8 | 50.6±24.7 | 0.5588 |
| Mean Physician's Global VAS (0–100) ± SD | 42.3±22.7 | 35.0±24.2 | 0.3880 |
| Rheumatoid factor positive, number (%) | 14 (53.9) | 4 (33.3) | 0.3067 |
| Anti-citrullinated protein antibody positive, number (%) | 13 (50.0) | 4 (33.3) | 0.4862 |
| Inflammatory back pain ever present, number (%) | 0 (0.0) | 3 (25.0) | 0.0261 |
| Psoriatic skin lesion ever present, number (%) | 5 (19.2) | 7 (58.3) | 0.0258 |
| Nail lesion present, number (%) | 4 (15.4) | 4 (33.3) | 0.2319 |
| Cluster: Group 1 (synovitis-dominant), number (%) | 12 (46.2) | 0.0 (0.0) | 0.0065 |

SD: standard deviation; IQR: interquartile range; SPARCC: Spondyloarthritis Research Consortium of Canada; HAQ-DI: Health Assessment Questionnaire-Damage Index; VAS: visual analogue scale; NSAID: non-steroidal anti-inflammatory drug.

over enthesitis (Fig 2D–2F). These groups did not clearly coincide with either clinical diagnosis (Fig 2B), ACPA positivity (Fig 2C), or rheumatoid factor positivity, illustrating again the heterogeneity of musculoskeletal inflammation within the same diagnosis or seropositivity.

Unexpectedly, there were no statistically significant differences in the numbers of any ILC populations in the peripheral blood between RA and SpA (Fig 3A). Instead, numbers of all ILC populations, particularly ILC3 and ILC1, were significantly higher in the synovitis-non-dominant group than in the synovitis-dominant group (Fig 3B). While ILCs are characterized by their tissue-resident roles [20], recent studies have suggested that ILC3 migrates from the gut to inflammatory sites through peripheral blood in SpA [24, 45]. Our data support the link between peripheral blood ILC and musculoskeletal inflammation not only in SpA but also in RA.

In our study, clustering of patients on the basis of 20 serum cytokine/chemokine levels did not clearly correspond to clinical diagnosis or ultrasound-based clustering (Fig 4). Among 11 individual cytokines/chemokines that were significantly different between RA and SpA, 10 were more elevated in the former (S1 Table), probably reflecting the higher inflammatory activity in RA patients than in SpA patients enrolled in this study. These cytokines/chemokines unexpectedly included IL-17A, IL-17F, and IL-23, which are considered to play more marked roles in SpA than in RA [46, 47]. Given that serum levels of these three cytokines were not significantly different between synovitis-dominant and synovitis-nondominant groups (S2 Table), the reason for this unexpected result might be the higher disease activity in RA and the presence of some synovitis-nondominant RA patients who had high serum levels of these cytokines/chemokines in our study. β-defensin 2, the only serum protein that was significantly

higher in SpA than in RA, was preferentially elevated in PsA. Indeed, the seven patients with the highest serum levels of β-defensin 2, a small cluster shown in Fig 4, were all PsA patients. This result confirms the already established role of β-defensin 2 as a potent serum biomarker of psoriasis [38, 48]. Collectively, these data indicate that the cytokines/chemokines measured in this study are generally not reliable biomarkers for peripheral musculoskeletal inflammation.

In our pilot, small scale analysis, the synovitis-dominant pattern was more significantly associated with clinical response to MTX than diagnosis. This result was expected as it is in line with the notion that MTX is efficacious for arthritis, which is usually a clinical manifestation of synovitis, but not for enthesitis in SpA [9]. However, the evidence related to this theory has been conflicting [49, 50]. One possible reason for the inconsistent results may be the limited or varying capability of physical examination to discriminate between arthritis, enthesitis, and dactylitis [15]. To support this, the predictive values of joint counts and enthesitis/dactylitis scores for MTX response in our study were much smaller compared with the ultrasound-based inflammation pattern (Table 2).

The limitations of this study include the small sample size, which limited the reliability and made performing multivariate analysis difficult. Second, the disease activity, especially in SpA, was not high, which might have influenced the results. However, it is technically difficult to match disease activity between two different diseases. Moreover, a fairly wide variety of active lesions were also present in SpA patients, which was sufficient for the distinct inflammation patterns to be illustrated with UMAP. Third, ultrasound was performed by multiple examiners. This probably has increased the heterogeneity of evaluation, while the heterogeneity can make the results more generalizable. Fourth, these sonographers were not completely blinded to the clinical information (i.e., they could have access if they intended to), which could have biased their evaluation.

In summary, our data demonstrate that patients with RA and SpA can be classified into two groups representing synovitis-dominant and synovitis-nondominant inflammation patterns. Our data also show that these patterns are associated with peripheral blood ILC counts and predictive of clinical response to MTX. These results indicate that imaging-detected patterns of musculoskeletal inflammation reflect underlying molecular/cellular pathophysiology and can be utilized to establish more individualized treatment for both RA and SpA.

## Supporting information

**S1 Fig. Gating method to identify innate lymphoid cells by flow cytometry.** After live lymphoid cells, singlets, and CD45[+] cells were gated, innate lymphoid cells (ILCs) were identified as lineage markers[−]CD127[+] cells. Lineage markers included CD3, CD14, CD19, CD11c, FcεRIα, CD94, CD123, CD34, and CD16. Group 1–3 ILCs (ILC1-3) were defined as CRTH2[−] c-kit[−]CD56[−] cells, CRTH2[+] cells, and CRTH2[−] c-kit[+] cells, respectively. FSC: forward scatter; SSC: side scatter.
(EPS)

**S2 Fig. Differences in total ultrasound scores between rheumatoid arthritis and spondyloarthritis.** Total synovitis/tenosynovitis/enthesitis scores are compared between patients with rheumatoid arthritis (RA) and those with spondyloarthritis (SpA).
(EPS)

**S1 Table. Differences in serum cytokine/chemokine levels between rheumatoid arthritis and spondyloarthritis.**
(DOCX)

**S2 Table. Differences in serum cytokine/chemokine levels between groups based on ultrasound.**
(DOCX)

**S3 Table. Correlations between serum cytokine/chemokine levels and numbers of peripheral innate lymphoid cell populations.**
(DOCX)

## Acknowledgments

We thank Ms. Kazumi Nemoto for helping with PBMC isolation and Ms. Chisato Ishii for data acquisition and collection. We would also like to thank the Chiba University Academic Link Center for assistance with language editing by a native English speaker.

## Author Contributions

**Conceptualization:** Manami Kato, Kei Ikeda.

**Data curation:** Manami Kato.

**Formal analysis:** Manami Kato, Kei Ikeda, Eiryo Kawakami.

**Investigation:** Manami Kato, Kei Ikeda, Takahiro Sugiyama, Kazuma Iida, Kensuke Suga, Nozomi Nishimura, Norihiro Mimura, Tadamichi Kasuya, Takashi Kumagai, Hiroki Furuya, Shunsuke Furuta, Kotaro Suzuki.

**Methodology:** Manami Kato, Kei Ikeda, Shigeru Tanaka, Akira Suto, Kotaro Suzuki, Eiryo Kawakami.

**Project administration:** Manami Kato, Kei Ikeda.

**Supervision:** Kei Ikeda, Hiroshi Nakajima.

**Validation:** Hiroshi Nakajima.

**Visualization:** Eiryo Kawakami.

**Writing – original draft:** Manami Kato, Kei Ikeda.

**Writing – review & editing:** Manami Kato, Kei Ikeda, Shigeru Tanaka, Taro Iwamoto, Arifumi Iwata, Shunsuke Furuta, Akira Suto, Kotaro Suzuki, Eiryo Kawakami, Hiroshi Nakajima.

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
