## [Decision Letter · Decision Letter 0]

5 May 2021

PONE-D-21-07558

Associations of ultrasound-based inflammation patterns with peripheral innate lymphoid cell populations, serum cytokines/chemokines, and treatment response to methotrexate in rheumatoid arthritis and spondyloarthritis

PLOS ONE

Dear Dr. Ikeda,

Thank you for submitting your manuscript to PLOS ONE. After careful consideration, we feel that it has merit but does not fully meet PLOS ONE’s publication criteria as it currently stands. Therefore, we invite you to submit a revised version of the manuscript that addresses the points raised during the review process.

Please revise this paper.

We look forward to receiving your revised manuscript.

Kind regards,

Gordana Ivanac

Academic Editor

PLOS ONE

Journal Requirements:

2. In your Methods section, please provide additional information about the participant recruitment method and the demographic details of your participants. Please ensure you have provided sufficient details to replicate the analyses such as descriptions of where participants were recruited and where the research took place.

3. Please provide a sample size and power calculation in the Methods, or discuss the reasons for not performing one before study initiation.

4. Please note that PLOS does not permit references to 'data not shown.' Authors should provide the relevant data within the manuscript, the Supporting Information files, or in a public repository. If the data are not a core part of the research study being presented, we ask that authors remove any references to these data.

Reviewers' comments:

Reviewer's Responses to Questions

**Comments to the Author**

1. Is the manuscript technically sound, and do the data support the conclusions?

Reviewer #1: Yes

Reviewer #2: Yes

2. Has the statistical analysis been performed appropriately and rigorously? 

Reviewer #1: Yes

Reviewer #2: I Don't Know

3. Have the authors made all data underlying the findings in their manuscript fully available?

Reviewer #1: Yes

Reviewer #2: Yes

4. Is the manuscript presented in an intelligible fashion and written in standard English?

Reviewer #1: Yes

Reviewer #2: Yes

5. Review Comments to the Author

Reviewer #1: The authors present an interesting study regarding association between musculoskeletal inflammation patterns with peripheral blood innate lymphoid cell (ILC) populations, serum cytokines/chemokines, and treatment response to methotrexate in patients with rheumatoid arthritis (RA) and 30spondyloarthritis (SpA). The study methodology seems sound and results and conclusions are in agreement with current knowledge, also bringing new information about ultrasound patterns occurring in muskuloskeletal inflammation. Although no association was found between the patterns and clinical diagnosis, I still find this data interesting and worth of publishing. My primary concern regarding the study is if there was only one person performing ultrasound exams or were there more examiners? Also, was there examiner blinding regarding the clinical diagnosis or not? Noting this in methodology section and entering short discussion about possible (dis)advantages of approach used in the study is crucial for better interpretation of the results. If this is added to the manuscript, I would firmly approve it for publishing.

Reviewer #2: In the Materials and methods you explain that you performed ultrasound examination for enthesitis at extensor digitorum insertions into 1st-5th middle phalanx, while in Figure 1 extensor digitorum insertion into proximal phalanx is shown.

Which site is correct? Could you name reference for evaluating this enthesis site?

6. PLOS authors have the option to publish the peer review history of their article (what does this mean?). If published, this will include your full peer review and any attached files.

Reviewer #1: **Yes: **Eugen Divjak

Reviewer #2: No

---

## [Author Response · Author response to Decision Letter 0]

6 May 2021

Please see "Response to Reviewers and Academic Editor" file.

---

## [Editor Report · Decision Letter 1]

11 May 2021

Associations of ultrasound-based inflammation patterns with peripheral innate lymphoid cell populations, serum cytokines/chemokines, and treatment response to methotrexate in rheumatoid arthritis and spondyloarthritis

PONE-D-21-07558R1

Dear Dr. Ikeda,

We’re pleased to inform you that your manuscript has been judged scientifically suitable for publication and will be formally accepted for publication once it meets all outstanding technical requirements.

Kind regards,

Gordana Ivanac

Academic Editor

PLOS ONE
---

## [Editor Report · Acceptance letter]

14 May 2021

PONE-D-21-07558R1 

Associations of ultrasound-based inflammation patterns with peripheral innate lymphoid cell populations, serum cytokines/chemokines, and treatment response to methotrexate in rheumatoid arthritis and spondyloarthritis 

Dear Dr. Ikeda:

I'm pleased to inform you that your manuscript has been deemed suitable for publication in PLOS ONE. Congratulations! Your manuscript is now with our production department. 

Kind regards, 

on behalf of

Dr. Gordana Ivanac 

Academic Editor

PLOS ONE